# Oncologic Outcomes of Laparoscopic Radical Hysterectomy Using the No-Look No-Touch Technique for Early Stage Cervical Cancer: A Propensity Score-Adjusted Analysis

**DOI:** 10.3390/cancers13236097

**Published:** 2021-12-03

**Authors:** Atsushi Fusegi, Hiroyuki Kanao, Naoki Ishizuka, Hidetaka Nomura, Yuji Tanaka, Makiko Omi, Yoichi Aoki, Tomoko Kurita, Mayu Yunokawa, Kohei Omatsu, Koji Matsuo, Naoyuki Miyasaka

**Affiliations:** 1Department of Gynecologic Oncology, Cancer Institute Hospital of the Japanese Foundation for Cancer Research, Tokyo 135-8550, Japan; atsushi.fusegi@jfcr.or.jp (A.F.); hidetaka.nomura@jfcr.or.jp (H.N.); yujit@belle.shiga-med.ac.jp (Y.T.); makiko.omi@jfcr.or.jp (M.O.); yoichi.aoki@jfcr.or.jp (Y.A.); t-kurita@med.uoeh-u.ac.jp (T.K.); mayu.yunokawa@jfcr.or.jp (M.Y.); kohei.omatsu@jfcr.or.jp (K.O.); 2Clinical Research Center, Cancer Institute Hospital of the Japanese Foundation for Cancer Research, Tokyo 135-8550, Japan; naoki.ishizuka@jfcr.or.jp; 3Division of Gynecologic Oncology, Department of Obstetrics and Gynecology, University of Southern California, Los Angeles, CA 90033, USA; koji.matsuo@med.usc.edu; 4Norris Comprehensive Cancer Center, University of Southern California, Los Angeles, CA 90033, USA; 5Department of Obstetrics and Gynecology, Tokyo Medical and Dental University, Tokyo 113-8510, Japan; n.miyasaka.gyne@tmd.ac.jp

**Keywords:** cervical cancer, laparoscopic surgery, minimally invasive surgery, radical hysterectomy, tumor spillage, no-look no-touch technique

## Abstract

**Simple Summary:**

Minimally invasive radical hysterectomy is contraindicated in early stage cervical cancer cases because of poor prognosis. The no-look no-touch technique (NLNT) eliminates intraoperative tumor spillage and may improve survival outcomes. We evaluated oncologic outcomes of laparoscopic radical hysterectomy performed using NLNT. We compared the outcomes of abdominal radical hysterectomy and NLNT using inverse probability of treatment weighting. We found no significant differences in disease-free survival between the groups, even in patients with tumor sizes ≥ 2 cm. We also studied NLNT’s non-inferiority to abdominal radical hysterectomy by evaluating heterogeneity between the results of the Laparoscopic Approach to Cervical Cancer (LACC) trial and those of our study. We did not observe significant heterogeneity, although there was a trend toward a lower hazard ratio in our study compared with the non-inferiority margin in the LACC trial. Laparoscopic radical hysterectomy using NLNT is a plausible alternative surgical treatment for early stage cervical cancer.

**Abstract:**

We evaluated oncologic outcomes of laparoscopic radical hysterectomy using the no-look no-touch technique (NLNT). We analyzed patients with early stage (IA2, IB1, and IIA1, FIGO2008) cervical cancer treated between December 2014 and December 2019. The primary endpoint was disease-free survival (DFS). We compared the outcomes of the abdominal radical hysterectomy (ARH) and NLNT groups using a Cox model with inverse probability of treatment weighting (IPTW), according to propensity scores. We also evaluated NLNT’s non-inferiority to ARH using an evaluation of heterogeneity between the results of the Laparoscopic Approach to Cervical Cancer (LACC) trial and our study. ARH and NLNT were performed in 118 and 113 patients, respectively. The median follow-up duration was 3.2 years. After IPTW adjustment, the 3-year DFS rates (NLNT 92.4%; ARH 94.0%) and overall survival rates did not differ significantly between the groups. Furthermore, the 3-year DFS rates for patients with tumor sizes ≥ 2 cm in the NLNT (85.0%) and ARH (90.3%) groups did not differ significantly. No significant heterogeneity was observed between the LACC trial and our study (I^2^ = 60.5%, *p* = 0.111), although there was a trend toward a lower hazard ratio in our study. Laparoscopic radical hysterectomy using NLNT provides a favorable prognosis for early stage cervical cancer.

## 1. Introduction

Radical hysterectomy is the standard surgical approach for women with early stage uterine cervical cancer and is associated with a 5-year disease-free survival (DFS) rate of >80% [1,2,3,4]. For decades, minimally invasive radical hysterectomy (MIRH), including laparoscopic surgery, was adopted to treat cervical cancer [5,6,7,8,9,10]. However, in 2018, the Laparoscopic Approach to Cervical Cancer (LACC) trial, a phase 3, multicenter, randomized study, revealed that MIRH was associated with poor prognosis compared with open abdominal radical hysterectomy (ARH); the risks of recurrence and death were four and six times higher, respectively [11]. Several subsequent large population analyses and a high-quality meta-analysis of observational research produced results corroborating the LACC trial [12,13,14,15,16,17,18]. ARH is now the recommended standard based on those findings, and MIRH is only performed in limited cases, including low-risk cervical cancer cases [1,2]. However, the exact cause of poor survival after MIRH remains unknown. Additionally, the studies mentioned above did not assess the quality control of the surgical procedure, making it difficult to interpret the results.

Intraoperative tumor spillage during laparoscopic surgery for various malignant tumors is highly concerning [19,20,21,22,23,24,25,26]. Among patients with cervical cancer, intraperitoneal tumor exposure during minimally invasive surgery was associated with a significantly poorer prognosis [26]. Therefore, tumor spillage owing to surgery-related factors (including tumor exposure under the pneumoperitoneum and intraoperative perforation or direct manipulation of the tumor) may cause cancer cell dissemination during laparoscopic surgery [27,28].

Previous studies, including our own, showed that intraoperative tumor spillage was a prime concern, and strategies aimed at limiting tumor interference when performing MIRH, including avoidance of tumor spillage using methods such as the no-look no-touch technique (NLNT) [29,30,31], were associated with improvements in cervical cancer DFS rates. However, these retrospective studies did not adjust for confounding factors. Interestingly, an observational study of 693 patients [18], which did include propensity score analysis, found that MIRH with protective vaginal closure resulted in oncologic outcomes similar to those of ARH (hazard ratio (HR): 0.63, 95% confidence interval (CI): 0.15–2.59). However, these data were limited, as only 43 participants underwent MIRH with a protective procedure. Altogether, whether the oncologic outcomes of patients undergoing MIRH using techniques to avoid tumor spillage are truly equivalent to those undergoing ARH, remains inconclusive.

This cohort study, with propensity score analysis, aimed to compare the survival outcomes of patients who underwent laparoscopic radical hysterectomy (LRH) for early stage cervical cancer using NLNT with those who underwent ARH. Moreover, we evaluated NLNT’s non-inferiority to ARH using the evaluation of heterogeneity between the results of the Laparoscopic Approach to Cervical Cancer (LACC) trial and our study.

## 2. Materials and Methods

### 2.1. Study Design and Endpoints

We conducted a cohort study to compare oncologic outcomes between patients who underwent ARH and LRH using NLNT for early stage cervical cancer between December 2014 and December 2019 at the Japanese Foundation for Cancer Research. The institutional review board at our institution approved this study (approval date: 21 July 2020, protocol no. 2020-1087), and all patients provided informed consent for this trial. We obtained patients’ clinical and pathological data by conducting chart reviews. Some patients in the current investigation were included in our previous study, where 163 patients who underwent radical hysterectomy between 2014 and 2017 in our hospital were evaluated [29].

The primary endpoint was DFS, defined as the time from surgery to detection of the first disease recurrence by imaging or death from any cause. The secondary endpoint was overall survival (OS), defined as the time from surgery to end for any reason. Data of patients with no evidence of recurrence or death were censored on the date of the last follow-up. Finally, we assessed the extent of heterogeneity between the LACC trial and our study to evaluate non-inferiority.

### 2.2. Participants

Patients who met the following criteria were included: (i) had previously untreated cervical cancer (those who underwent diagnostic conization could be included); (ii) were at clinical stages IA2, IB1, and IIA1, based on the revised 2008 International Federation of Gynecology and Obstetrics staging system; (iii) had histologically confirmed cervical cancer, including squamous cell carcinoma, adenocarcinoma, and adenosquamous carcinoma; and (iv) underwent type III radical hysterectomy and pelvic lymphadenectomy (per the Piver–Rutledge–Smith classification) performed by board-certified gynecologic oncologists.

Patients underwent either ARH or LRH, based on their choice, following preoperative consultation and discussions regarding their diagnosis with gynecologists. All NLNT procedures were conducted by a single board-certified surgeon using the same protocols (H.K.). We excluded patients who underwent LRH without NLNT and those who received preoperative radiation therapy, concurrent chemoradiation therapy, or neoadjuvant chemotherapy.

### 2.3. Treatment Procedures and Follow-Up

Procedures, outcomes (including the extent of resection), and postoperative course of LRH using NLNT have been described previously [29]. NLNT comprises four steps aimed at preventing tumor cell spillage: (i) creation of the vaginal cuff (Figure 1A), (ii) manipulation of the uterus without inserting an intra-uterine manipulator (Figure 1B), (iii) minimal handling of the uterine cervix, and (iv) bagging the specimen.

The need for adjuvant therapy was based on pathological findings, as follows: (i) patients with intermediate risks (i.e., lymphovascular involvement, tumors with a large diameter of ≥4 cm, and/or ≥50% myometrial invasion) received chemotherapy; (ii) patients with high risks (i.e., positive surgical margins and/or ≥3 metastatic lymph nodes) received concurrent chemoradiation therapy; and (iii) patients with 1–2 lymph node metastases and/or paracolpium invasion received chemotherapy alone. However, two patients in the ARH group received adjuvant radiotherapy according to their preference and/or at the discretion of their treating physician.

After completing the treatment, routine follow-up examinations were performed as per the clinical standard of practice: a pelvic examination was conducted, a tumor marker profile was obtained every 3 months, and computed tomography was performed annually.

### 2.4. Patient-Related Variables

We compared the following variables between patients who underwent ARH and LRH using NLNT: age, body mass index (kg/m^2^), history of diagnostic conization, histological subtype, tumor diameter, presence of vaginal tumor extension, operative time, estimated blood loss, intraoperative and postoperative complications, length of hospital stay, pathological tumor stage, lymphovascular invasion, tumor invasion depth, surgical margin status, type of adjuvant therapy, DFS, OS, and anatomical location of recurrence. Intraoperative complications were defined as an injury to abdominal organs, including the bladder, ureter, intestines, great vessels, and nerves. Postoperative complications were defined as events occurring up to postoperative day 30 that were classified as grade ≥2 (i.e., requiring pharmacological treatment other than grade 1 complications), according to the Clavien–Dindo classification [32].

### 2.5. Statistical Analyses

We compared continuous variables using the Mann–Whitney U test and categorical variables using Pearson’s chi-squared test. Moreover, we used the inverse probability of treatment weighting (IPTW) method to adjust for confounding variables. IPTW method is a statistical technique that uses propensity scores to calculate statistics standardized to a group different from the original cohort; it helps obtain unbiased estimates of treatment effect. Individual propensities for LRH using NLNT were calculated using a multivariable logistic regression model that included age, body mass index, history of diagnostic conization, histological subtype, preoperative tumor diameter, and presence of vaginal tumor extension as explanatory variables. These factors comprised all clinical information that could influence our preoperative decision making, regarding surgical procedures. Data for each factor were available for all patients; therefore, there was no evidence of selection bias when performing the IPTW method.

The 3- and 4.5-year DFS rates were estimated using the Kaplan–Meier method, and 95% CIs for the primary outcome were calculated using the Greenwood formula. Survival curves were generated using the Kaplan–Meier method. Cox proportional hazard models with or without IPTW, according to the propensity score, were used to estimate the HR and 95% CIs for the treatment effect on DFS. Log-rank tests with and without IPTW were used to compare the groups. The IPTW method was used for the primary analysis. We performed subgroup DFS analyses according to tumor size (≥2 cm).

We evaluated the heterogeneity between the LACC trial and our study using the I^2^ statistic in which non-inferiority was judged based on whether the upper limit of the 95% CI was lower than the margin. Two investigators (N.I. and N.K.) independently screened the statistical analyses for quality control.

We used SAS software, version 9.4 (SAS Institute Inc., Cary, NC, USA), for survival analysis and EZR software, version 1.41 (Saitama Medical Center, Jichi Medical University, Saitama, Japan), for all other statistical analyses. All reported statistical tests were two-sided, with a significance level of 0.05.

## 3. Results

### 3.1. Patient Characteristics

Among 231 Japanese patients with early stage cervical cancer, 118 underwent ARH, and 113 underwent LRH using NLNT. The median follow-up time was 3.2 years. Patient clinicopathologic characteristics are presented in Table 1. Patients were significantly younger in the NLNT group (median, 42.0 years) than in the ARH group (46.5 years; *p* = 0.001). Patients had smaller clinical tumor diameters in the NLNT group (median, 1.9 cm) than in the ARH group (2.5 cm; *p* = 0.013). Tumor stromal invasion (stromal invasion depth ≥1/2) rates were lower in the NLNT group than in the ARH group (37/113 (32.7%) vs. 60/118 (48.7%) patients; *p* = 0.008). Except for the type of adjuvant therapy administered (*p* = 0.002), the remaining characteristics did not significantly differ between the groups. Among patients with tumor size ≥ 2 cm in the NLNT and ARH groups, there were significant differences in age (42.0 vs. 47.0 years; *p* = 0.004) and tumor-stromal invasion rate (26/56 (46.4%) vs. 56/81 (69.1%) patients; *p* = 0.013), while there were no significant differences in clinical tumor diameters (2.8 vs. 2.9 cm) and adjuvant therapy type (*p* = 0.082; (Appendix A)).

### 3.2. Efficacy

The 3- and 4.5-year DFS rates did not significantly differ between the NLNT and ARH groups (3-year DFS rate: 95.9% vs. 91.4%; 4.5-year DFS rate: 93.8% vs. 88.3%; HR: 0.522 (95% CI: 0.181–1.510; *p* = 0.223); Figure 2A). This difference between the groups was maintained even after adjustment using the IPTW method, indicating well-balanced covariates between the two groups (3-year DFS rate: 92.4% vs. 94.0%; 4.5-year DFS rate: 90.9% vs. 91.4%; HR: 1.283 (95% CI: 0.462–3.566; *p* = 0.841); Figure 2B). The 3- and 4.5-year OS rates did not significantly differ between the two groups (3-year OS rate: 100.0% vs. 98.2%; 4.5-year OS rate: 100.0% vs. 96.2%; *p* = 0.138; the HR could not be calculated because there was no event in the NLNT group).

### 3.3. Subgroup Analysis

We further analyzed differences in DFS among patients with tumor sizes ≥ 2 cm. Four (7.1%) and 11 (13.6%) patients in the NLNT and ARH groups, respectively, experienced relapse; the 3- or 4.5-year DFS rates did not significantly differ between the NLNT and ARH groups (3-year DFS rate: 91.5% vs. 87.7%; 4.5-year DFS rate: 91.5% vs. 83.6%; HR: 0.581 (95% CI: 0.184–1.833; *p* = 0.349); Figure 2C). This difference was maintained after IPTW adjustment (3-year DFS rate: 85.0% vs. 90.3%; 4.5-year DFS rate: 85.0% vs. 86.8%; HR: 1.155 (95% CI: 0.390–3.418; *p* = 0.780); Figure 2D).

### 3.4. Comparison with the Non-Inferiority Margin in the LACC Trial

We assessed the heterogeneity between the results of the LACC trial and those of our study (Figure 3). In the LACC trial, the HR of minimally invasive surgery to open surgery for disease recurrence or death owing to cervical cancer was 3.74 (95% CI: 1.63–8.58), which was not inferior according to the predefined non-inferiority margin of 1.791 that was converted from the percentage point of −7.2 of the DFS rate. Significant heterogeneity between the two studies was not observed (I^2^ = 60.5%; *p* = 0.111). The upper limit of the 95% CI of the HR in all patients and patients with tumor size ≥ 2 cm was not lower than the non-inferiority margin in the LACC trial. The HR of patients with tumor sizes < 2 cm could not be calculated because the ARH group had no relapses.

### 3.5. Perioperative Characteristics

Patient perioperative characteristics are presented in Table 2. The NLNT group had significantly shorter operation times (median, 293 vs. 375 min; *p* < 0.001), less blood loss (120 vs. 490 mL; *p* < 0.001), and shorter hospital stays (14 vs. 18 days; *p* < 0.001) than the ARH group. There were no significant differences in intraoperative or postoperative complications between the groups: one patient in each group had an intraoperative complication (both were bladder injuries); nine in the ARH group had postoperative complications (peritonitis [*n* = 4], ileus [*n* = 1], delayed urinary tract injury [*n* = 1], pulmonary embolism [*n* = 1], symptomatic lymphatic cyst [*n* = 1], and delayed colon perforation [*n* = 1]); and four in the NLNT group had postoperative complications (peritonitis [*n* = 2], ileus [*n* = 1], and delayed urinary tract injury [*n* = 1]). No conversions occurred in the NLNT group.

Regarding the anatomical location of the first recurrences in patients in the ARH group, relapses were observed at the vaginal stump [*n* = 2], vaginal mucosa [*n* = 1], regional lymph nodes [*n* = 3], distant lymph node [*n* = 1], lungs [*n* = 2], omentum [*n* = 1], and spine [*n* = 1]. On the other hand, in the NLNT group, the patients experienced relapse at the vaginal stump [*n* = 2], regional lymph nodes [*n* = 2], pelvic cavity [*n* = 1], and lungs [*n* = 2] (Table 3).

## 4. Discussion

In this cohort study, we compared oncologic outcomes between patients who underwent ARH and LRH, using NLNT for early stage cervical cancer. Our data revealed no difference in survival outcomes between patients who underwent ARH and LRH using NLNT. Conversely, our results did not show non-inferiority similar to the LACC trial results. Additionally, heterogeneity was not significant between the LACC trial and our study, although I^2^ was moderately large. This result suggested that the proper prognosis caused by MIRH shown in the LACC trial may not apply to all MIRHs.

Although worse outcomes have been reported following MIRH than those following ARH [11,12,13,14,15,16,17], there is almost no information regarding surgery-specific factors associated with poor prognosis. The reason for the poor prognosis was thought to be the surgical technique, including tumor spillage in the surgical field. The concept of NLNT is to prevent the potential risks of the procedure, which are (i) creation of the vaginal cuff, (ii) manipulation of the uterus without inserting an intra-uterine manipulator, (iii) minimal handling of the uterine cervix, and (iv) bagging the specimen. In some studies, including NLNT, the outcomes of patients who underwent MIRH with precautions against tumor spillage did not differ from those of patients who underwent ARH [18,29,30,31].

However, besides caveats, including small sample sizes and retrospective designs, the studies had a selection bias that caused an imbalance in patient distribution between the ARH and MIRH groups. Since the publication of the LACC trial, patients preoperatively determined to have low-risk cervical cancer were generally assigned to the MIRH group. Our study overcame these limitations by including a large sample size and incorporating propensity score adjustments and prospective observations, based on the definitive follow-up protocol. Under these conditions, the oncologic outcomes between the two groups were similar. Therefore, our study provides evidence that patient prognosis following LRH using NLNT is comparable to that following ARH.

One of the strengths of our study is that all surgical procedures in both groups were type III radical hysterectomy and pelvic lymphadenectomy, and all NLNT procedures were performed by the same board-certified surgeon using the same protocols. This guaranteed quality control regarding the surgical procedure and surgeon’s skill. Moreover, instead of radiation or chemoradiation, which is considered effective in reducing pelvic recurrence, chemotherapy was administered as an adjuvant treatment in the NLNT group. Our data showed that an improved prognosis could be achieved without irradiation for local control by avoiding tumor spillage.

Notably, our study showed that the DFS in the NLNT and ARH groups was comparable, even among patients with tumor size ≥2 cm. Differences in patient prognosis according to tumor size (<2 vs. ≥2 cm) have been previously reported. The differences are recognized in the most recent classification of the International Federation of Gynecology and Obstetrics [33,34,35]. Moreover, MIRH was associated with acceptable oncologic outcomes in patients with tumor sizes < 2 cm [36,37,38]. Therefore, oncologic outcomes in patients who underwent LRH with protection against tumor spillage, including those with tumor sizes ≥ 2 cm, are important. We found no significant differences between the two groups for the type of adjuvant therapy administered when considering such patients. Therefore, we could compare oncologic outcomes without considering the effect of adjuvant therapy. Survival analysis of patients with tumor sizes ≥ 2 cm, adjusted using the IPTW method, revealed no significant difference in DFS between groups. Additionally, the HR was comparable to that of all patients in the study. However, caution is required in the interpretation of these findings, especially regarding the subgroup with tumor sizes ≥ 2 cm; our study suggested that NLNT can achieve a similar prognosis to ARH.

Unlike previous studies [11,39], we found that locoregional recurrence was not common in patients undergoing laparoscopy, which is plausible given the concept of NLNT. Previous Norwegian studies highlighted cancer cell insufflation due to anastomotic leakage as a possible cause of inferior outcomes of transanal total mesorectal resection in patients with rectal cancer [24], which may also be true for cervical cancer. Moreover, surgery-related factors, including using a uterine manipulator, direct contact with the tumor, and tumor exposure to the intraoperative field, may increase the risk of tumor spillage and result in poor prognosis [19,20,21,22,23,25,26,27]. These findings remain controversial but should nevertheless be noted. Our study provides evidence that LRH using NLNT can overcome these risks and reduce local recurrence rates.

We demonstrated the surgical feasibility of NLNT in a larger sample, and the surgical outcomes (shorter operative time, less blood loss, shorter hospital stay, and no significant difference in the intraoperative/postoperative complication profile) were consistent with those previously reported [29]. The reason for extended hospital stays in both groups compared with those reported in studies from other countries may be partly due to Japanese hospital protocols, wherein the patient is admitted 2 days before surgery and the urinary catheter is removed a minimum of 7 days after surgery.

Based on previously published data reporting that LRH using NLNT has favorable oncologic outcomes, we evaluated whether LRH using NLNT was non-inferior to ARH. We used the non-inferiority margin in the LACC trial as a reference and assessed the heterogeneity of the results of the LACC trial and those of our study. We did not observe significant heterogeneity between these results, although there was a trend toward lower HR in our study than the non-inferiority margin in the LACC trial. We posit that this was caused by insufficient statistical power; therefore, a larger sample size, in which the number of relapses was equivalent to that reported in the LACC trial, would provide a more precise assessment. This trend suggested that the poorer prognosis following MIRH indicated in the LACC trial may not apply to all procedures directly, and careful interpretation is necessary. Therefore, longer follow-up studies with precise evaluations of minimally invasive surgery, while preventing tumor spillage, are warranted.

Our study has some limitations. First, while the IPTW method can reduce bias regarding observed differences between groups, propensity score-based adjustments are still subject to biases from unobserved confounding factors. Second, our study comprised a limited number of patients with a short follow-up interval and was performed at a single center by a single surgeon, which may limit the generalizability of our findings. Third, our research may have exaggerated the treatment effect while providing insufficient statistical power to analyze the heterogeneity between the LACC trial and our study. Fourth, the differences in tumor stromal invasion may have influenced the survival analysis, resulting in overestimation of NLNT outcome. Finally, the differences between using or forgoing NLNT among patients who underwent LRH were not evaluated.

## 5. Conclusions

A favorable oncologic prognosis with feasible surgical outcomes was observed in the NLNT group, suggesting that LRH using NLNT is a plausible alternative surgical approach to treat early stage cervical cancer. Further studies are needed to evaluate the safety and feasibility of MIRH involving protective techniques to avoid intraoperative tumor spillage.

## Figures and Tables

**Figure 1 cancers-13-06097-f001:**
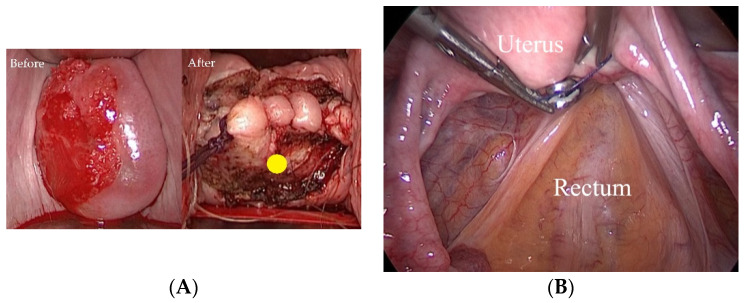
Concept of the “no-look no-touch” technique (**A**) before (left) and after (right) the creation of the vaginal cuff. Before the laparoscopic procedure, we created a cuff to isolate the tumor from the operative field. We placed a trocar (marked with a yellow circle) to prevent it from touching the tumor. (**B**) Manipulation of the uterus without insertion of a uterine manipulator. We use the forceps through the trocar placed in the posterior vaginal fornix to manipulate the uterus by handling the thread around the uterine body.

**Figure 2 cancers-13-06097-f002:**
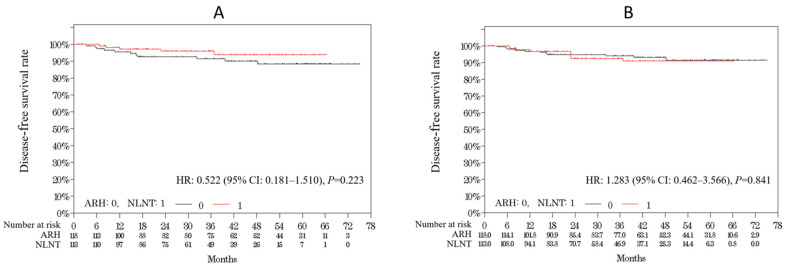
Kaplan–Meier survival analyses in patients with early stage cervical cancer who underwent ARH vs. LRH using NLNT. (**A**) Raw data of disease-free survival for all patients. (**B**) IPTW-adjusted data of disease-free survival for all patients. (**C**) Raw data of disease-free survival for patients with tumor size > 2 cm in diameter. (**D**) IPTW-adjusted data of disease-free survival for patients with tumor size > 2 cm in diameter. ARH—abdominal radical hysterectomy; NLNT—laparoscopic radical hysterectomy using no-look no-touch technique; IPTW—inverse probability of propensity score weighted; HR—hazard ratio; CI—confidence interval.

**Figure 3 cancers-13-06097-f003:**
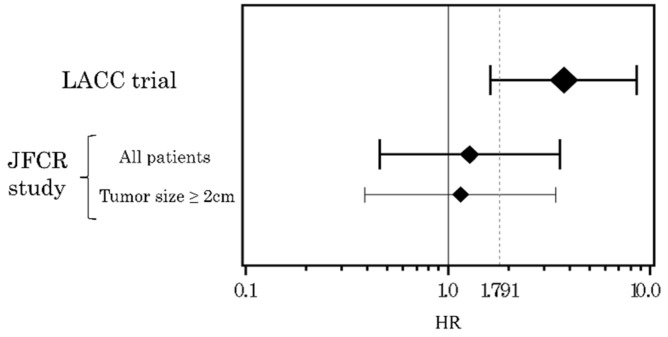
Comparison of heterogeneity between patients in the LACC trial and those in our study; hazard ratios with 95% confidence intervals for relapse among patients who underwent minimally invasive surgery (compared with patients who underwent open surgery). The size of the diamond is proportional to the inverse of the square of the standard error. The vertical dotted line is equivalent to the non-inferiority margin used in the LACC trial converted to the hazard ratio, which has a value of 1.791. The hazard ratio in patients with tumor size < 2 cm in diameter could not be calculated because no relapses occurred among patients in the ARH group. LACC—Laparoscopic Approach to Cervical Cancer (trial); JFCR—Japanese Foundation for Cancer Research; NLNT—laparoscopic radical hysterectomy using the no-look no-touch technique; HR—hazard ratio.

**Table 1 cancers-13-06097-t001:** Clinicopathologic characteristics of all patients.

Patient Characteristics	ARH (*n* = 118)	NLNT (*n* = 113)	*p*-Value
Age (years)	46.5 (41–58.5)	42.0 (37–48.0)	0.001
BMI (kg/m^2^)	21.45 (20.1–23.8)	21.20 (19.2–23.6)	0.201
Tumor size (cm)	2.5 (1.5–3.0)	1.9 (0.0–2.8)	0.013
Vaginal invasion			0.085
Negative	105 (88.9%)	108 (95.6%)	
Positive	13 (10.1%)	5 (4.4%)	
Histological subtype			0.247
SCC	57 (48.3%)	55 (48.7%)	
AC	33 (28.0%)	40 (35.4%)	
ASC	28 (23.7%)	18 (15.9%)	
Post-conization			0.733
No	98 (83.1%)	91 (80.5%)	
Yes	20 (16.9%)	22 (19.5%)	
pT			0.113
1a1	0	1 (0.9%)	
1a2	1 (0.8%)	2 (1.8%)	
1b1	83 (70.3%)	88 (77.9%)	
1b2	10 (8.5%)	3 (2.7%)	
2a1	10 (8.5%)	7 (6.1%)	
2a2	2 (1.7%)	6 (5.3%)	
2b	12 (10.2%)	6 (5.3%)	
pN			0.316
N0	93 (78.8%)	95 (84.1%)	
N1	25 (21.2%)	18 (15.9%)	
Stromal invasion			0.008
<1/2	58 (51.3%)	76 (67.3%)	
≥1/2	60 (48.7%)	37 (32.7%)	
Parametrium invasion			0.316
Negative	107 (90.7%)	107 (94.7%)	
Positive	11 (9.3%)	6 (5.3%)	
Venous invasion			0.56
Negative	87 (73.7%)	79 (69.9%)	
Positive	31 (26.3%)	34 (30.1%)	
Lymphatic invasion			0.79
Negative	66 (55.5%)	66 (58.4%)	
Positive	52 (44.5%)	47 (41.6%)	
Cut margin			0.066
Negative	111 (94.1%)	112 (99.1%)	
Positive	7 (5.9%)	1 (0.9%)	
Adjuvant therapy			0.002
None	55 (46.6%)	63 (55.7%)	
Radiation	2 (1.7%)	0	
Chemotherapy	48 (40.7%)	49 (43.4%)	
CCRT	13 (11.0%)	1 (0.9%)	

Data are presented as medians (interquartile ranges) or *N* (%); ARH—abdominal radical hysterectomy; NLNT—laparoscopic radical hysterectomy using no-look no-touch technique; BMI—body mass index; SCC—squamous cell carcinoma; AC—adenocarcinoma; ASC—adenosquamous carcinoma; CCRT—concurrent chemoradiation therapy.

**Table 2 cancers-13-06097-t002:** Perioperative characteristics.

Perioperative Characteristics	ARH (*n* = 118)	NLNT (*n* = 113)	*p*-Value
Operative time (min)	375 (330–458.5)	293 (260–330)	<0.001
Blood loss (mL)	490 (320–840)	120 (75–210)	<0.001
Intraoperative complications			1.000
Yes	1 (0.8%)	1 (0.9%)	
No	117 (99.2%)	112 (99.1%)	
Postoperative complications			0.254
Yes	9 (7.6%)	4 (3.5%)	
No	109 (92.4%)	109 (96.5%)	
Hospital stay (days)	18 (16–21)	14 (13–16)	<0.001

Data are presented as medians (interquartile ranges) or *N* (%), ARH, abdominal radical hysterectomy; NLNT—laparoscopic radical hysterectomy using the no-look no-touch technique.

**Table 3 cancers-13-06097-t003:** Characteristics of patients with relapse.

Patient	Procedure	Tumor Size (cm)	Histology	pTNM	Adjuvant Therapy	Recurrent Site	DFS (Months)
1	ARH	2.9	ASC	pT1bN0M0	None	Vaginal stump	3.3
2	ARH	3.5	SCC	pT1b1N1M0	Chemotherapy	Spine	5.6
3	ARH	2.1	ASC	pT1b1N1M0	Chemotherapy	Omentum	6.1
4	ARH	3.7	SCC	pT2a1N1M0	Chemotherapy	PLN	8.1
5	ARH	6.5	ASC	pT1b2N1M0	Chemotherapy	PLN	10.9
6	ARH	3.2	ASC	pT2bN1M0	Chemotherapy	PLN, PAN	15.1
7	ARH	4.6	SCC	pT1b2N1M0	Chemotherapy	Vagina	16.5
8	ARH	4.7	SCC	pT1b1N0M0	None	Vaginal stump	16.8
9	ARH	4.4	AC	pT2bN1M0	CCRT	Distant LN	32.7
10	ARH	5.0	ASC	pT2bN1M0	CCRT	Lung	40.2
11	ARH	2.5	AC	pT1b1N0M0	None	Lung	48.9
12	NLNT	2.2	ASC	pT1b1N0M0	None	Vaginal stump	6.7
13	NLNT	4.7	SCC	pT2bN1M0	Chemotherapy	Vaginal stump, PLN	8.5
14	NLNT	4.5	ASC	pT1b2N1M0	Chemotherapy	Lung	12.0
15	NLNT	4.0	ACC	pT2a2N0M0	None	Lung	23.2
16	NLNT	1.5	AC	pT1b1N0M0	None	Pelvic cavity, PAN	37.4

ARH—abdominal radical hysterectomy; NLNT—laparoscopic radical hysterectomy using no-look no-touch technique; SCC—squamous cell carcinoma; AC—adenocarcinoma; ASC—adenosquamous carcinoma; CCRT—concurrent chemoradiation therapy; PLN—pelvic lymph node; PAN—paraaortic lymph node; LN—lymph node; DFS—disease-free survival.

## Data Availability

The data presented in this study are available on request from the corresponding author.

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
