# Peer review of "Oncologic Outcomes of Laparoscopic Radical Hysterectomy Using the No-Look No-Touch Technique for Early Stage Cervical Cancer: A Propensity Score-Adjusted Analysis"

_cancers, 2021, doi:10.3390/cancers13236097_

Round 1
Reviewer 1 Report
This paper deals with a much debated and controversial topic: that is, the non-inferiority of minimally invasive surgery compared with laparotomic surgery in the treatment of early stage cervical cancer. Although this is a retrospective study, the propensity score-adjusted analysis is well conducted and the patient sample is sufficiently large. The presentation of the contents of the study is clear both in the materials and methods and in the discussion sections.
The strengths of the study are the statistical analysis, the long follow up and the large number of patients treated in a reproducible manner, always with the same surgical technique (type III radical hysterectomy). The standardization of the surgical technique with the creation of the vaginal cuff to avoid spillage in each single procedure is also commendable
Some critical points emerge from the tables:
1) A type III radical hysterectomy with lymphadenectomy was performed in all cases. Table 1 shows that in 4 cases the FIGO stage was between 1a1 and 1a2. Although it is a small percentage, this is still an overtreatment.
2) Table 1 demonstrates that in 10.2% of cases of ARH and 5.3% of cases of NLNT the stage of the disease was 2B, which represents a contraindication to surgery. Was this a clinical and instrumental staging error?
The authors argue that with the necessary precautions and the right surgical technique, such as the creation of a vaginal cuff (a strategy also adopted in some European surgical schools), the non-manipulation of the tumor during surgery, and specimen bagging, the minimally invasive procedure can be considered safe resulting in oncological outcomes that are comparable to the open technique, contrary to what is stated by the LACC trial. This is undoubtedly acceptable
The authors then analyzed the survival of all patients with T> 2 cm and concluded that NLNT can achieve a prognosis similar to ARH, even for patients with tumor size ≥2 cm (line 308 to 311). In this study, the mean tumor diameter in NLNT cases is <2 cm (exactly 1.9 cm) against 2.5 cm in ARH cases (table 1). Although these data are not statistically significant, the authors' conclusions should be supported by more data, and above all, the average diameter of the tumor of patients treated with the minimally invasive technique should be greater than, and not less than 2 cm.
This study is well conducted and demonstrates that for T ≤ 2 cm in diameter the minimally invasive approach to cervical cancer is safe and effective. This is in line with the consensus of the scientific community which recommends minimally invasive technique only for T <2.
We do not yet have sufficient data to extend the indications to tumors with a diameter greater than 2 cm, whose intervention with minimally invasive technique could present an untoward risk today.
This paper is suitable for publication after minor revision.
Author Response
Thank you very much for spending your time on reviewing our manuscript and giving us good advice. We agree with all your advice and have incorporated them into our revised manuscript. The revisions corresponding to your comments and suggestions are written in red font.
Point 1: Some critical points emerge from the tables:
1) A type III radical hysterectomy with lymphadenectomy was performed in all cases. Table 1 shows that in 4 cases the FIGO stage was between 1a1 and 1a2. Although it is a small percentage, this is still an overtreatment.
2) Table 1 demonstrates that in 10.2% of cases of ARH and 5.3% of cases of NLNT the stage of the disease was 2B, which represents a contraindication to surgery. Was this a clinical and instrumental staging error?
Response 1:
We believe that the discrepancy between preoperative diagnosis and postoperative histopathological diagnosis (both understaging and upstaging) is a problem of preoperative assessment.
However, we also believe that our preoperative positive diagnosis rate was high compared to that of previous reports (cf. Zhang W, et al. Staging early cervical cancer in China: data from a multicenter collaborative. Int J Gynecol Cancer. 2019 May 15:ijgc-2019-000263. doi: 10.1136/ijgc-2019-000263.)
It is desirable to continue improving diagnostic approaches to achieve accurate preoperative diagnosis in the future.
Point 2: The authors then analyzed the survival of all patients with T> 2 cm and concluded that NLNT can achieve a prognosis similar to ARH, even for patients with tumor size > 2 cm (line 308 to 311). In this study, the mean tumor diameter in NLNT cases is <2 c m (exactly 1.9 cm) against 2.5 cm in ARH cases (table 1). Although these data are not statistically significant, the authors' conclusions should be supported by more data, and above all, the average diameter of the tumor of patients treated with the minimally invasive technique should be greater than, and not less than 2 cm.
Response 2:
We agree with your comment. As per your comment, we have restructured the expression about the survival analysis of all patients with T> 2 cm (Page 10. Lines 316–320, red color).
We believe that incorporating your advice into this revision has made the manuscript better. Thank you once again.

Reviewer 2 Report
The methodological section of the present study is questionable. This study included 231 women in total. However, it is unclear how many women were evaluated following the propensity score comparison. Furthermore, the propensity score comparison should include a sample with no significant differences between the independent variables. This is in order to make the two study groups (NLNT versus ARH) homogeneous and comparable. All of this is missing in this study. There is no sample size calculation. If 231 women were included in the total, with the propensity score they will be even less, so the study loses power. I don't think it adds much to current knowledge.
Author Response
Thank you very much for spending your time on reviewing our manuscript and giving us good advice. We agree with all your advice and have incorporated them into our revised manuscript. The revisions corresponding to your comments and suggestions are written in blue font.
Point 1: The methodological section of the present study is questionable. This study included 231 women in total. However, it is unclear how many women were evaluated following the propensity score comparison.
Response 1:
You advised us that the sample size calculation after propensity score adjustment should be described. We understand that if we used “propensity score matching,” we have to describe the sample size calculation after propensity score adjustment. However, in this manuscript, we used “the inverse probability of treatment weighting (IPTW) method,” in which we used the entire cohort, and each individual in the cohort was assigned a weight based on the likelihood of exposure to the treatment effect under investigation. There seems to be a misunderstanding here.
Our description was also unclear; thus, we have added a new explanation (page 4. Lines 154–157, blue color) to make it easier for the readers to understand.
We are sincerely apologetic for the confusion.
Point 2: The propensity score comparison should include a sample with no significant differences between the independent variables. This is in order to make the two study groups (NLNT versus ARH) homogeneous and comparable
Response 2:
We agree with you. We chose variables (age, body mass index, history of diagnostic conization, histological subtype, preoperative tumor diameter, and presence of vaginal tumor extension) to calculate the propensity score. All these factors were important preoperative variables for clinical decision making in choosing a surgical procedure between open and NLNT radical hysterectomy. However, we did not choose these variables as they had significant differences between the two study groups.
We believe that incorporating your advice into this revision has made the manuscript better. Thank you once again.

Reviewer 3 Report
In the echo of the LACC Trial and its consequents in terms of re-thinking, re-planning and re- organizing every day gynaecological practice for surgical cervical cancer treatment, the essential everyday need for studies like this one becomes even more crucial.
The question arises is how and for how long in the future are we going to be able to achieve such a goal. The study period of this study (12/2014-12/2019) has given that opportunity. However, clinical practice in the present and future suggest the opposite. How is going to draw prospectively “unethical” trial?
The study by Atsushi Fusegi et al. aims to compare the survival outcomes of patients who underwent laparoscopic radical hysterectomy (LRH) for early-stage cervical cancer using NLNT with those who underwent ARH. Moreover, the authors evaluated no-look no-touch technique’s (NLNT) non-inferiority to open abdominal radical hysterectomy (ARH) by using the evaluation of heterogeneity between the results of the Laparoscopic Approach to Cervical Cancer (LACC) trial and their study.
The authors included in their analysis patients who: (i) had previously untreated cervical cancer including even those who underwent diagnostic conization prior to the operation; (ii) Patients who were at clinical stages of IA2, IB1, and IIA1 based on the revised 2008 International Federation of Gynecology and Obstetrics staging system; (iii) patients that had histologically confirmed cervical cancer, including squamous cell carcinoma, adenocarcinoma, and adenosquamous carcinoma; and (iv) patients who underwent type III radical hysterectomy and pelvic lymphadenectomy (per the Piver-Rutledge-Smith classification) performed by board certified gynecologic oncologists.
However, the authors denote that the choice of treatment for either ARH or LRH, was based on patients’ choice, following preoperative consultation and discussions regarding their diagnosis with gynaecologists.
I am afraid that the particular “concept” of treatment option choice is the major limitation of the study, in terms of “blindness” regarding the extension of the disease and this is confirmed if we look in detail the characteristics of the patients’ distribution among the 2 groups, and remains strong enough limitation for misleading conclusions.
As the authors denote, patients were significantly younger in the NLNT group (median, 42.0 years) than in the ARH group (46.5 years and additionally had smaller clinical tumor diameters in the NLNT group (median, 1.9 cm) than in the ARH group (2.5 cm; P=0.013) Furthermore, tumor stromal invasion (stromal invasion depth ≥1/2) rates were lower in the NLNT group than in the ARH group.
As they conclude that despite the “obvious differences” mentioned above, the 3- and 4.5-year DFS rates did not significantly differ between the NLNT and ARH groups (3-year DFS rate: 95.9% vs. 91.4%; 4.5-year DFS rate: 93.8% vs. 88.3%; HR: 0.522 [95% CI: 0.181–1.510; P=0.223]; even after adjustment by IPTW method, real life indicates that such a conclusion continues to rise questions in terms of “ethical issues” in the echo of the LACC trial results.
Of course, and in spite of the strengthen of the particular study, that all surgical procedures in both groups were type Ⅲ radical hysterectomies and pelvic lymphadenectomies, and all NLNT procedures were performed by the same board-certified surgeon who used the same protocols, real life suggests that this may happen in only few advanced gynaecological centres around the world.
However, and despite the fact that all the above limitations and possible mis-perfections, are not strong enough to allow decision for rejection of the particular very well written and presented study. My impression is that the particular study should be considered for publication in your prestigious journal of Cancers and similar publications by other scientific groups in the particular field should also be encouraged in the near future.
Please upgrade the discussion
Author Response
Thank you very much for spending your time on reviewing our manuscript and giving us good advice. We agree with all your advice and have incorporated them into our revised manuscript. The revisions corresponding to your comments and suggestions are written in green font.
Point 1: The question arises is how and for how long in the future are we going to be able to achieve such a goal. The study period of this study (12/2014-12/2019) has given that opportunity. However, clinical practice in the present and future suggest the opposite. How is going to draw prospectively “unethical” trial?
Response 1:
The conclusions of the LACC trial could be considered a signal against minimally invasive surgery for early-stage cervical cancer. However, the trial could not provide an explanation for the poor prognosis of minimally invasive surgery (MIS).
The SUCCOR study and our report suggested that intraoperative avoidance of tumor spillage may contribute to improved prognosis. We further added the evaluation of heterogeneity in this study, considering the possibility that the MIS in this study might vary from surgeries performed in the LACC study. Although the results were not significant, we believe that the conclusions of the LACC study may not apply to all MIS, including the no-look no-touch technique.
Since the LACC study is an RCT, we need to take its results seriously. However, we also believe that we have an obligation to verify the results by conducting a high-quality prospective "observational" study.
Point 2: I am afraid that the particular “concept” of treatment option choice is the major limitation of the study, in terms of “blindness” regarding the extension of the disease and this is confirmed if we look in detail the characteristics of the patients’ distribution among the 2 groups, and remains strong enough limitation for misleading conclusions.
As the authors denote, patients were significantly younger in the NLNT group (median, 42.0 years) than in the ARH group (46.5 years and additionally had smaller clinical tumor diameters in the NLNT group (median, 1.9 cm) than in the ARH group (2.5 cm; P=0.013) Furthermore, tumor stromal invasion (stromal invasion depth ≥1/2) rates were lower in the NLNT group than in the ARH group.
As they conclude that despite the “obvious differences” mentioned above, the 3- and 4.5-year DFS rates did not significantly differ between the NLNT and ARH groups (3-year DFS rate: 95.9% vs. 91.4%; 4.5-year DFS rate: 93.8% vs. 88.3%; HR: 0.522 [95% CI: 0.181–1.510; P=0.223]; even after adjustment by IPTW method, real life indicates that such a conclusion continues to rise questions in terms of “ethical issues” in the echo of the LACC trial results.
Response 2:
Since this study used a propensity score, which was measured preoperatively, we believe that the effect of confounding factors has been eliminated as much as possible. Age and tumor size were among these factors, and their prognostic impact calculated using the IPTW method was less than it might appear.
As for the difference in tumor stromal invasion observed as an outcome rather than a preoperative factor, we think that this could be a major limitation of the study, as Reviewer 3 mentioned, because it could directly affect the survival analysis. In this regard, we have described the limitation caused by the differences in tumor stromal invasion (page 10, lines 356–357, green color).
We believe that incorporating your advice into this revision has made the manuscript better. Thank you once again.

Reviewer 4 Report
The report by Fusegi et al. suggests that No look not touch technique is a remedy for side effects of laparoscopic route in radical hysterectomy. The authors used this methods for second time (first time in 2019 report) and didn't report this fact. According to me the above should be stressed in the report. Additionally they report: "some patients in the current investigation were included in our previous study". How many ? Which ones? What was the system of patients recruitment in current study?
Author Response
Thank you very much for spending your time on reviewing our manuscript and giving us good advice. We agree with all your advice and have incorporated them into our revised manuscript. The revisions corresponding to your comments and suggestions are written in purple font.
Point 1: The report by Fusegi et al. suggests that No look not touch technique is a remedy for side effects of laparoscopic route in radical hysterectomy. The authors used this methods for second time (first time in 2019 report) and didn't report this fact. According to me the above should be stressed in the report.
Response 1:
We agree with you. Intraoperative tumor spillage during laparoscopic radical hysterectomy is a prime concern, and one of the techniques to overcome this problem is the “no-look no-touch technique.” We have emphasized this fact in the Introduction (page 2, lines 68–69, purple color).
Point 2: They report: "some patients in the current investigation were included in our previous study". How many? Which ones? What was the system of patients recruitment in current study?
Response 2:
To clarify, we have added this information to the Materials and Methods (page 2, lines 93–94, purple color). Despite the impact of the LACC trial, the patient inclusion criteria have not changed significantly. We described the patients’ recruitment in the Materials and Methods (page 3, lines 101–115 of the revision version).
We believe that incorporating your advice into this revision has made the manuscript better. Thank you once again.
